# In Vitro Antifungal Activity of Selected Essential Oils against Drug-Resistant Clinical *Aspergillus* spp. Strains

**DOI:** 10.3390/molecules28217259

**Published:** 2023-10-25

**Authors:** Valeria Allizond, Lorenza Cavallo, Janira Roana, Narcisa Mandras, Anna Maria Cuffini, Vivian Tullio, Giuliana Banche

**Affiliations:** Department Public Health and Pediatrics, Microbiology Division, University of Torino, 10126 Turin, Italy; valeria.allizond@unito.it (V.A.); lorenza.cavallo@unito.it (L.C.); janira.roana@unito.it (J.R.); annamaria.cuffini@unito.it (A.M.C.); vivian.tullio@unito.it (V.T.); giuliana.banche@unito.it (G.B.)

**Keywords:** *Aspergillus*, essential oils, clove essential oil, lemongrass essential oil, geranium essential oil, antifungal activity

## Abstract

Background: Treatment options for aspergillosis include amphotericin B (AMB) and azole compounds, such as itraconazole (ITZ). However, serious side effects related to these antifungal agents are increasingly evident, and resistance continues to increase. Currently, a new trend in drug discovery to overcome this problem is represented by natural products from plants, or their extracts. Particularly, there is a great interest in essential oils (EOs) recognized for their antimicrobial role towards bacteria, fungi and viruses. Methods: In this study, we evaluated the antifungal activity of eleven commercial EOs—clove, eucalyptus, geranium, hybrid lavender, lavender, lemon, lemongrass, neroli, oregano, tea tree and red red thyme—in comparison with AMB and ITZ against *Aspergillus flavus*, *A. fumigatus* and *A. niger* clinical isolates. Antifungal activity was determined by broth microdilution method, agar diffusion technique, fungistatic and fungicidal activities and vapor contact assay. Results: Gas chromatography–mass spectrometry analysis displayed two groups of distinct biosynthetical origin: monoterpenes dominated the chemical composition of the most oils. Only two aromatic compounds (eugenol 78.91% and eugenyl acetate 11.64%) have been identified as major components in clove EO. Lemongrass EO exhibits the strongest antimicrobial activity with a minimum inhibitory concentration of 0.56 mg/mL and a minimum fungicidal concentration of 2.25–4.5 mg/mL against *Aspergillus* spp. strains. Clove and geranium EOs were fairly effective in inhibiting *Aspergillus* spp. growth. Conclusions: These results demonstrate the antimicrobial potential of some EOs and support the research of new alternatives or complementary therapies based on EOs.

## 1. Introduction

Filamentous fungi as *Aspergillus genus* are ubiquitous, versatile and saprophytic ascomycetes, so they can cause a wide range of clinical manifestations. *Aspergillus* conidia are also widespread in the environment, and for people that come into contact and then retain conidia, a pattern of clinically significant outcomes may occur ranging from asymptomatic colonization to invasive infection [1,2].

Aspergilli can cause hypersensitivity reactions such as allergic sinusitis and allergic bronchopulmonary aspergillosis, chronic pulmonary aspergillosis, superficial and cutaneous mycoses, such as subungual onychomycosis, and otomycosis [2]. Furthermore, as we now know, invasive aspergillosis (IA), a life-threatening condition, continues to be the most problematic disease with high levels of morbidity and mortality, especially in patients with compromised immunity in terms of disease management. *Aspergillus fumigatus* causes a majority of infections, followed by *A. flavus*, *A. niger* and *A. terreus* [2,3].

Three classes of antifungal agents are available for the treatment of aspergillosis including polyenes, azoles and echinocandins. However, several examples of aspergillosis are difficult to treat and relatively recurrent. This is because the use of antimycotic drugs available at present, most of which are fungistatics, is limited by the emergence of toxicity, poor solubility, new resistant strains and low efficacy. Additionally, some antifungals such as azoles can sometimes cause allergic reactions or other side effects [4,5]. 

Natural antifungal agents could provide alternative solutions with fewer or no side effects. Thus, their combined use with standard therapeutic agents could improve the efficacy of treatment [6]. Essential oils (EOs) are well-known antimicrobials because they possess antifungal, antibacterial, anti-inflammatory and antioxidant properties [7,8]. 

The antimicrobial properties of EOs derive mainly from their composition, the quantity of the dominant compounds and the method used to test their effectiveness. Due to their lipophilic nature, EOs and their active components penetrate and accumulate easily in the lipid bilayer of cytoplasmic membranes [9,10,11]. 

The antibacterial properties of EOs have been observed in several studies [8,12]. The main target of the EOs is the cell membrane or the cell wall. The membrane loses its integrity while its permeability and fluidity are increased [6,13]. This interrupts many cellular activities (such as energy production, membrane transport). In particular, EOs containing a high percentage of phenolic compounds, such as carvacrol, eugenol and thymol, have a greater ability to alter the permeability and function of cell membrane proteins, by penetrating the phospholipid layer of the bacterial wall, blocking normal functions of proteins. Gram-positive bacteria are more sensitive to EOs than Gram-negatives, due to the complexity of their outer membrane, which acts as a barrier to cell membrane permeability [10,12]. 

EO extracts from many plants such as basil, citrus, fennel, lemongrass, oregano, rosemary and thyme have shown remarkable antifungal activity against many fungal pathogens. Against fungi (yeasts and molds), the action of EOs is often similar to that demonstrated against bacteria. EOs have the ability to penetrate and destroy the fungal cell wall and the cytoplasmic membranes through a permeabilization process. This leads to the disintegration of mitochondria membranes. The permeabilization process is caused by alterations in the flow of electrons that are observed in the electron transport system. This could also damage lipids, proteins and nucleic acid contents of cells infected by fungi [6,11,12]. 

In addition, their use as preservatives, disinfectants or therapeutic agents has been under intensive research. In particular, EOs from *Origanum vulgare* L. (oregano oil), *Thymus vulgaris* L. (red red thymeoil) and *Lavandula angustifolia* Mill. (lavender oil) (*Lamiaceae* family), clove EO from *Eugenia caryophyllata* L. (*Myrtaceae*) and geranium EO from *Pelargonium graveolens* L’Hèrin.exAit. *(Geraniaceae)* have been used for centuries for their medicinal properties [5,7,10,14].

The aim of this study was to evaluate the antifungal activity of eleven different EOs chosen to investigate their in vitro activity against *A. flavus*, *A. fumigatus* and *A*. *niger.* Therefore, the goal was to increase the amount of data for the efficacy of EOs against these molds, because there is little information when compared with their efficacy against bacteria and yeasts.

## 2. Results 

### 2.1. Chemical Component Analysis

The major components of the EOs used were determined by gas chromatography–mass spectrometry: eugenol (78.91% *v*/*v*), eugenyl acetate (11.64% *v*/*v*), β-caryophyllene (6.04% *v*/*v*), α-humulene (0.69% *v*/*v*), α-copaene (0.27% *v*/*v*) in clove; 1.8 cineole (80.64% *v*/*v*), limonene (10.41% *v*/*v*), α-pinene (2.52% *v*/*v*), p-cymene (3.65% *v*/*v*), α-phellandral (0.49% *v*/*v*) in eucalyptus; citronellal + neral (33.22% *v*/*v*), geraniol-formate (15.56% *v*/*v*), iso-menthone (5.71% *v*/*v*), linalol (4.19% *v*/*v*) in geranium; linalol (33.83% *v*/*v*), linalyl acetate (27.43% *v*/*v*), canphor (7.10% *v*/*v*), 1,8 cineole (4.84% *v*/*v*), borneol (4.18% *v*/*v*) in hybrid lavender; linalol (27.11% *v*/*v*), linalyl acetate (24.4% *v*/*v*), β-ocimene (9.78% *v*/*v*), caryophyllene (5.36% *v*/*v*), 4-terpineol (5.11% *v*/*v*) in lavender; limonene (69.25% *v*/*v*), β-pinene (11.37% *v*/*v*), γ-terpinen (7.86% *v*/*v*), sabinene (1.98% *v*/*v*), α-pinene (1.75% *v*/*v*) in lemon; geraniol (22.64% *v*/*v*), limonene (7.74% *v*/*v*), canphene (7.66% *v*/*v*), methyl isoeugenol (6.81% *v*/*v*), geranyl acetate (5.90% *v*/*v*) in lemongrass; linalol (31.49% *v*/*v*), β-pinene (17.32% *v*/*v*), limonene (15.76% *v*/*v*), β-ocimene (6.67% *v*/*v*), linalyl acetate + geraniol (4.04% *v*/*v*) in neroli; carcacrol (62.61% *v*/*v*), ρ-cymene (12.36% *v*/*v*), γ-terpinen (3–9% *v*/*v*), thymol (0.5–5% *v*/*v*), β-caryophyllene (0.5–4% *v*/*v*) in oregano; thymol (26.5% *v*/*v*), ρ-cymene (16.2% *v*/*v*), limonene (13.2% *v*/*v*), α-pinene (11.5% *v*/*v*), carvacrol (7.8% *v*/*v*), eugenol (0.1–0.5% *v*/*v*) in red red thyme; terpinen-4-ol (35.88% *v*/*v*), ϒ-terpinen (19.65% *v*/*v*), α-terpinen (8.64% *v*/*v*), ρ-cymene (4.61% *v*/*v*), 1.8 cineole (4.07% *v*/*v*) in tea tree. EOs used and their major components are reported in Appendix A.

### 2.2. Determination of Antifungal Activity

Table 1 shows the family, scientific name and common name of the EOs used.

Table 2 shows MIC’s values obtained using AMB and ITZ, and 11 EOs used against the *Aspergillus* spp. clinical strains. Based on susceptibility testing results, *Aspergillus* spp. strains were found to be resistant to both the reference antifungal drugs. One strain of *A. fumigatus* (MOL 3628) and two strains of *A. niger* (MOL 5015 and AdS217) showed MIC values of 2 µg/mL towards ITZ, while all other strains show MIC values > 2 µg/mL towards azole. As far as AMB is concerned, three strains of *A. niger* (MOL 121973, Ads 121822 and AdS217) showed MIC values = 2 µg/mL.

Amongst EOs, lemongrass EO exhibited a high inhibitory activity on the growth of all *Aspergillus* spp. strains, with low MICs ranging from 0.56 to 1.12 mg/mL. In particular, this oil showed good efficacy towards *A. flavus*, 5 strains *of A. fumigatus* and *A. niger* AdS 121822 (MIC = 0.56 mg/mL) (Table 2). Clove and geranium EOs were fairly effective in inhibiting *Aspergillus* spp. growth, displaying MICs ranging from 0.56 to 2.25 mg/mL and from 0.56 to 4.5 mg/mL, respectively, with greater efficacy against *A. fumigatus* and *A. niger*. Lemon EO showed weak antifungal activity against most of the strains studied. Eucalyptus, hybrid lavender, lavender, neroli, oregano, tea tree and red thyme EOs, conversely, were not effective in inhibiting fungal growth with MIC values of 4.5–9 mg/mL (Table 2). Generally, MFCs were one or more concentrations higher than MICs, suggesting a fungicidal activity of the EOs at low concentrations (Table 3). MFC values for clove EO ranging between 2.25–9 mg/mL, geranium EO between 1.12–4.5 mg/mL and lemongrass EO values between 2.25–4.5 mg/mL. 

Results of the inhibitory activity in agar medium of reference drugs and EOs, that were most effective with reference to MIC values, are reported in Table 4 and Figure 1, Figure 2 and Figure 3. Values are expressed as halo of inhibition in millimeters. All tested strains showed greater resistance to AMB, with inhibition haloes of no more than 13 mm. In contrast, ITZ-related inhibition halos for *A. fumigatus* were higher, with values between 17–25 mm, while for only one strain of *A. niger* the zone of inhibition was 14 mm (Table 4). As regards agar disc diffusion method in presence of EOs, the data showed lower values at 50 µg than at 150 µg. The maximum antimycotic activity was shown by clove EO against *A. fumigatus* and *A. niger* strains (Table 4, Figure 2 and Figure 3) at both concentrations (50 and 150 µg): statistical difference in antifungal activity were found between clove 75% (150 µg) and AMB for all *Aspergillus* spp. (*p* < 0.05), while EO significantly increased the inhibition zone diameter only in *A. niger* compared to ITZ (*p* < 0.05). Geranium and lemongrass EOs exhibited no activity at 50 µg against *A. niger.* Only *A. niger* 777 was inhibited by lemongrass OE at 150 µg (Table 4, Figure 3). Among all the oils tested, clove at 150 µg showed significant differences in antimycotic activity against *A. niger* and *A. flavus* compared to geranium at 50 and 150 µg (*p* < 0.05), and against *A. flavus* in comparison with lemongrass (*p* < 0.05).

The determination of fungistatic and fungicidal activities on *Aspergillus* spp. strains is presented in Figure 4. The absence of growth after transferring the fungal section to sterile medium indicates fungicidal properties. Conversely, growth has a fungistatic effect. The fungicidal activities were evident in particular for clove and geranium EOs against all strains tested (Figure 4). On three strains of *A. fumigatus* tested (BEN 177045, data not showed, MOL 3628 and MOL 113549), clove, geranium and lemongrass EOs were fungicides at a concentration of 0.125% (1.12 mg/mL). Only lemongrass EO showed fungistatic activity at concentrations of 0.06% (0.56 mg/mL) on *A. flavus* AdS 5834 and of 0.125% (1.12 mg/mL) on *A. niger* MOL 121973. 

Figure 5 and Figure 6 show EOs activity in the vapor phase. All tested EOs did not inhibit fungal growth at the 5% concentration. In the presence of clove and geranium EOs, only *A. niger* AdS 777 (Figure 6B) showed limited and not widespread growth in the plate. Conversely, at the 75% concentration clove, geranium and lemongrass EOs inhibited the growth of all strains assayed except *A. niger* MOL 121973, AdS 121822 (Figure 6C,D) and *A. fumigatus* AdS 5822, MOL 113549 (Figure 5A,C) in the presence of clove EO. 

## 3. Discussion

Many plant-derived natural compounds and EOs have exhibited great potential against fungal propagation. Our interest has focused on *Aspergillus* spp., which cause a spectrum of syndromes depending on the degree of immunosuppression in the host. Treatments for aspergillosis today are affected by an increasing number of species that have developed resistance [15]. As a consequence, there is more and more attention in investigating effects, mechanisms of action and interactions of natural products and their phytocomplexes that have been in empirical use for centuries. Natural products could prove to be excellent supportive drugs to traditional ones. As now known, EOs are complex natural mixtures which can contain a large number of components at quite different concentrations, so much so that they are considered a potential source of bioactive antimicrobial compounds to improve antifungal treatment, because it is unthinkable that a fungus can acquire resistance to all components [13,16]. The literature increasingly documents the antimicrobial activity of Eos, including geranium, lavender, lemongrass, oregano, thyme and tea tree [8,17,18,19,20,21]. 

In this research, all the oils tested exhibited different degrees of antifungal activity against *Aspergillus* strains. Not all oils tested have shown good antifungal activity. The maximum antimycotic activity was shown by *C. nardus* (lemongrass oil), followed by *E. caryophyllata* (clove oil) and *P. graveolens* (geranium oil) (Table 2). 

Among *Cymbopogon* spp. EOs, the antifungal activity of *C. citratus*, commonly named lemongrass, is the most often reported in the literature [22]. Boukhatem et al. reported antimycotic activity in liquid and vapor phases of *C. citratus* on yeast and filamentous fungi strains including *A. flavus, A. fumigatus* and *A. niger.* Lemongrass EO appears to be a potentially valuable antifungal and anti-inflammatory agent for the treatment of acute inflammatory skin diseases, and the antifungal property of lemongrass EO could be due to two major monoterpene aldehydes (geranial and neral) [22]. We have analyzed a different species, *C. nardus*, and our results indicate that the activity of lemongrass EO is presumably due to the presence of citronellal and geraniol (61.64% of the total composition of the EO). Similarly, de Billerbeck et al. (2001) found that *C. nardus* EO (citronellal was 42% and geraniol 22.64% *v*/*v*) resulted in complete growth inhibition of *A. niger* at 800 mg/L on agar plates [23]. Their results showed that *C. nardus* oil tested on *A. niger* appears to be more toxic than *C. citratus* oil tested on the plasma membrane of fungus; in the presence of *C. nardus* EO at 200 mg/L, it was seen to be irregular and associated with the formation of lomasomes. Bansod et al. [24] reported that activity of *C. citratus* (lemongrass oil) and *E. caryophyllata* (clove oil) inhibited *A. fumigatus* and *A. niger* at 0.25% *v*/*v* (*billerbech*). 

In our study, we also found that EOs extracted from *P. graveolens* (geranium oil) and *Eugenia caryophyllata* (clove oil) demonstrated strong antifungal activity on all the species of *Aspergillus* tested (Table 2). Clove was the second most effective EO, which was able to inhibit all fungal strains at concentrations of 0.06–0.25% (*v*/*v*) (0.056–2.25 mg/mL) (Table 2). The results are different from those reported by Pinto et al. (2009) in which MIC values, on an environmental isolated *A. fumigatus* strain, ranged from 0.32–0.64% (*v*/*v*) [17]. The higher MIC values found by Pinto et al. may be due to the different characteristics between clinical and environmental strains. 

Studies conducted by Horváth et al. demonstrated that MIC results for lemongrass, clove and thyme EOs were lower against *A. fumigatus* than against *A. terreus*. In fact, MIC values for lemongrass EO against *A. fumigatus* are 0.78 mg/mL, MIC values for clove EO against *A. fumigatus* and *A.terreus* are 1.56 mg/mL and 3.13 mg/mL, respectively, while MIC values for thyme EO against *A. fumigatus* and *A. terreus* are 0.39 mg/mL and 0.78 mg/mL, respectively [6]. Our values are lower for some strains, even though in the study of Horváth et al. the fungi were, like ours, isolated from human infections. Furthermore, the antifungal efficacy of clove EO has also been documented by other authors on different candida species [25,26,27]. The antimycotic activity of clove EO due to the presence of eugenol (78.91% *v*/*v*) is well known. The main antimicrobial mechanism of eugenol is the increase in cell membrane permeability and therefore the dissolution of the fungal cell [17,28,29,30,31].

The EO of geranium was found to have strong antimycotic activity against *A. fumigatus* and *A. niger* (Table 2). The literature provides little information on the efficacy of this EO, but the antifungal activity of the oil and its main components, citronellol and geraniol, are well-known in literature. A combination therapy of oils, or their active components, and synthetic antibiotics could be particularly useful against candida and aspergilli. Indeed, some authors have demonstrated that a combination (such as geraniol, as well as citronellol) may be useful in clinical situations requiring the use of AMB or ketoconazole [32,33]. The oils of *E. globosum*, *L. x hybrid*, *L. officinalis*, *C. aurantium, O. vulgare, M. alternifolia* and *T. vulgaris* were ineffective against aspergilli (MIC > 1% *v*/*v* corresponding to 9 mg/mL, Table 2). These data are not reflected in literature. Hossain et al. (2019) reported the efficacy of thyme and oregano EOs against *Aspergillus* spp. From the American Type Culture Collection (ATCC). The combination treatment of these EOs displayed higher activity than the individual oregano or thyme treatments [34]. In our study, the lack of efficacy may be due to various factors (species taxonomy, climate, OE extraction method, concentration of active compounds, fungal genus and concentration of inoculum) and whether the strain is clinical, environmental or an ATCC strain [35,36]. 

Oregano EO showed higher values, ranging from 0.5% (4.5 mg/mL) to >1% *v*/*v* (9 mg/mL) (Table 2), reflecting a lower efficacy on *Aspergillus* spp. strains than those reported by Carmo et al. (2008), where MICs ranged from 0.08–0.02% (*v*/*v*). The fungal strains used by the authors were environmental strains and not clinical strains, which may help to explain the lower efficacy of our data [37]. 

Sakkas et al. (2017) reported that thyme EO is one of the most potent antifungals out of all oils tested against different *Aspergillus* spp., including *A. flavus* [10]. The antimicrobial effect of thyme oil is attributed to carvacrol and thymol. Its antimicrobial spectrum is broad and includes only bacteria and yeasts [7,10].

Therefore, comparison of the data obtained in this study with previously published results is problematic. In our study, the components include two groups of distinct biosynthetical origin (monoterpenes dominated the chemical composition of the most oils). These terpenes are formed from the coupling of two isoprene units (C10) [13]. Linalool was the main component identified in the hybrid lavender, lavender and neroli EOs. Only two aromatic compounds (eugenol 78.91% and eugenyl acetate 11.64%) have been identified as major components in clove EO. The antimicrobial activity of EOs and their components are influenced by the composition of plant oils, local climatic and environmental conditions and experimental conditions [24,25]. Indeed, although CLSI methods are accepted for determining the in vitro susceptibility of microorganisms to antimicrobial agents, it is debatable how EOs, or even fungi, fit into these protocols [38]. The need for a standard and reproducible method for the evaluation of oils is increasingly underlined by various authors [24]. 

Thus, researchers have proposed several techniques to elucidate the mechanism used via EOs to inhibit or inactivate the fungal cells [9]. Most of the in vitro studies performed tend to evaluate MIC and MIC/MFC of the volatile compound or extract and the probable mechanism of action at the basis of the antifungal effect [39]. The results obtained with the MIC study generally confirm those obtained in the disc diffusion assay and support the antifungal activities of the oils against *Aspergillus* spp. Compared to AMB and ITZ, EOs tested exhibited a major inhibitory effect against aspergilli (Figure 2 and Figure 3). Geranium and lemongrass EOs, at the 25% tested concentration, show lower inhibition halos in comparison to clove (Table 4). The results may be due to a lower diffusion ability of these EO in the agar. Clove oil turned out to be the best inhibitor overall thanks to eugenol, which contributes most to its bioactivity [7]. 

The hydrophobic nature of most EOs prevents the uniform diffusion of these substances through the agar medium. In addition, other properties may have an affect: these include difficulty preparing the inoculum and differences in microbial growth. This method proposes qualitative and preliminary data [8,13,24]. Studies have been conducted to evaluate the utility of disc diffusion for filamentous fungi. The correlation with broth microdilution is not very good, and there are no interpretive criteria by which to evaluate the results. CLSI has developed a standard for testing. Disc spreading interpretive criteria for filamentous fungi are not available [40]. Furthermore, as to disagreement over the criteria for determining the endpoint of the inhibition zone diameters for azoles, it was measured at 80%. However, CLSI disc diffusion breakpoints were used for interpretive breakpoints [41].

In our study, we have evaluated the antifungal activity of OEs by direct contact tests. These results are very different from the direct contact assay, where clove, geranium and lemongrass EOs were successful in inhibiting the growth of the tested strains. In the presence of concentrations of EOs equal to the MIC, fungistatic or fungicidal effects were also demonstrated with the EO-enriched medium test (Figure 4). Based on the MBC/MIC ratio, if this ratio is ≤4, the effect is considered bactericidal/fungicidal, but if the MBC/MIC ratio > 4, the effect is defined as bacteriostatic/fungistatic [42]. Only lemongrass EO showed fungistatic activity at concentrations of 0.06% *v*/*v* (0.56 mg/mL) on *A. flavus* AdS 5834 and 0.125% *v*/*v* (1.12 mg/mL) on *A. niger* MOL 121973 (Table 2 and Table 3, Figure 4). Boukhatem et al. point out that lemongrass EO in the vapor phase is an effective antifungal system. This oil has several advantages over the liquid phase (higher potency at lower doses for the same effect) [22]. Our previous study against clinical filamentous fungi proved that clove oil possesses high antimicrobial activity in vapor phase [7,25]. The investigations of the effects of the use of EOs in the vapor state are still few and often, as in this case, differences in study methods can modify the efficacy results. Despite the differences between the methods, our results demonstrate that some EOs are very active on aspergilli.

## 4. Material and Methods

### 4.1. Strains

A total of 13 clinical *Aspergillus* strains were obtained from University Hospital Città della Salute e della Scienza di Torino hospitalized patients. These included *A. flavus* (AdS 5834) isolated from sputum, four *A. fumigatus* strains (AdS 5822, MOL 1517, BEN 177045, MOL 3628) isolated from bronchoalveolar lavage (BAL), three *A. fumigatus* strains (MOL 4470, MOL 4697, MOL 113549) isolated from sputum and five *A. niger* strains (AdS 777, MOL 121973, AdS 121822, MOL 5015, AdS 217) isolated from BAL. All isolates were identified by macroscopic and microscopic examination and stored in Sabouraud dextrose broth (Oxoid, Italy) with 20% glycerol at −80 °C. Before the experiments, the fungi were transferred to Potato Dextrose Agar (PDA, Oxoid, Segrate, Italy) fresh media and incubated for 7 days at 25 °C.

### 4.2. Antifungal Agents

The antifungal drug powders (≥98% purity by HPLC), such as amphotericin B (AMB) and Itraconazole (ITZ), were purchased from Sigma-Aldrich (Italy), dissolved in 100% dimethylsulfoxide (DMSO; Sigma-Aldrich) and stored at −20 °C until use. Standard antibiotics AMB and ITZ were used in order to control the sensitivity of the tested fungus. 

### 4.3. Essential Oils and Their Components

Commercial EOs of clove (*Eugenia caryophyllata* Trumb.), eucalyptus (*Eucalyptus globosus* Labill.), geranium (*Pelargonium graveolens* L’Hérit.exAit.), hybrid lavender (*Lavandula x hybrid*), lavender (*Lavandula officinalis* P.), lemon (*Citrus lemon* L.), lemongrass (*Cymbopogon nardus* L.), neroli (*Citrus aurantium* var. bitter L.), oregano (*Origanum vulgare* L.), tea tree (*Melaleuca alternifolia* Cheel.) and red thyme (*Thymus vulgaris* L.) (Table 1) were obtained by hydrodistillation and provided from Flora s.r.l. (Pisa, Italy). The EOs were analysed by GC-MS with a Perkin Elmer Clarus 500 gas chromatograph (Perkin Elmer, Milan, Italy), at PRIMAVERA LIFE GmbH. The producer guaranteed the chemical composition of each EO. The EOs were stored in amber glass vials.

### 4.4. Inoculum Preparation

Strains were subcultured in darkness at 30 °C on PDA. After a week, the conidial suspension was collected with sterile swabs and transferred to a flask containing NaCl 0.85%. Afterward, the number of conidia in suspension was counted before being diluted to reach an inoculum concentration of 2 × 10^4^ conidia/mL by a Neubauer chamber and adjusted by spectrophotometer (Thermo Scientific Genesys 20) at 675 nm. The resulting suspensions were confirmed by colony counts in triplicate on Sabouraud dextrose agar (SDA, Oxoid, Segrate, Italy) medium [7].

### 4.5. Broth Micro Dilution Method

In vitro antifungal potential activity of EOs, AMB and ITZ was tested according to the Clinical and Laboratory Standards Institute (CLSI M38-A2) microdilution reference method, with appropriate methodological changes for EOs [17,18,19,43]. The preparation of EO stock solutions was performed according to the previously mentioned procedure [18]. Briefly, each EO was dissolved in ethanol (1:2.5) and diluted 1:20 in RPMI-1640 medium (Sigma, Milan, Italy), buffered to pH 7.0 with 0.165 M 3-(*N*-morpholino) propane sulfonic acid (MOPS, Sigma-Aldric) and supplemented with 0.2% glucose to obtain a final concentration of 2% (*v*/*v*). Tween 80 (Sigma-Aldrich), with a final concentration of 0.5% *v*/*v*, was used to enhance the EOs’ solubility, without any inhibitory effect on mold growth. Two-fold dilutions of AMB and ITZ (range 64–0.12 µg/mL) and EOs (range 18–0.035 mg/mL) were prepared in a 96-well microtiter plate in RPMI-1640 with MOPS. After the addition of 100 µL of inoculum (2 × 10^4^ conidia/mL) and final concentrations of drugs (32–0.06 µg/mL) and/or EOs (ranged from 9 to 0.017 mg/mL) with Tween 80, plates were incubated at 30 °C for 7 days. RPMI-1640 with MOPS medium was used as growth control [7,19]. The minimum inhibitory concentration (MIC) of drugs and EOs is defined as the lowest concentration of the compound that causes a specified reduction in visible growth of the microorganism in microdilution wells as detected by the naked eye [41]. To evaluate the minimal fungicidal concentration (MFC) of drugs and EOs, 10 µL of sample was taken from each well showing no visual growth after incubation, spotted onto SDA and incubated at 30 °C for 72 h. The MFC was considered to be the lowest concentration that completely inhibited fungal growth [7,19].

### 4.6. Agar Diffusion Technique

The antifungal activity of AMB and ITZ was evaluated by Agar Disc Diffusion Technique. Briefly, the assay was performed in 90 mm diameter Petri dishes containing Mueller Hinton Agar (MHA, Becton Dickinson, Sparks, MD, USA) for drugs and on Malt Extract Agar (MEA, Oxoid, Italy) medium for EOs, to a depth of 4 mm. Different concentrations for assay, i.e., 20 μg/disc for AMB and 10 μg/disc for ITZ, were used. The drugs were dissolved in DMSO to obtained a 1000 μg/mL stock solution, from which the appropriate amount was drawn with a sterile pipette to achieve the concentration for each drug. EOs were screened for their antifungal activity by MIC of oil inhibiting the visible growth of aspergilli. Selected EOs were prepared at 25% and 75% concentrations by dissolving the oil in ethanol and emulsifying with 0.5% Tween 80. Sterile paper discs were soaked with 20 μL of each EO concentration [8,24,44]. The fungal inocula (2 × 10^6^ conidia/mL), previously prepared, were applied uniformly on the surface of MHA or MEA medium and spread by using a sterile glass loop. A negative control with only DMSO was set up in the disc to exclude solvent activity. Plates were incubated at 35 °C for 5 days. After incubation, the diameters of the growth inhibition zones were measured in millimeters to the nearest 0.1 mm using electronic calipers. The data of all the parameters were statistically analyzed. 

### 4.7. Determination of Fungistatic and Fungicidal Activities

To determine whether a given EO possessed a fungistatic effect and/or fungicidal activity, the procedure reported by Puškárová et al. [8] was used. For treatment sets, the EOs were incorporated into the molten MEA according to MIC values and mixed well. Then, the medium was poured into Petri dishes. In parallel, a positive control was set up with EO-free MEA. The inoculum was represented by a mycelia disc ranging 6 × 6 mm cut from the periphery of seven-day-old culture of aspergillus, which was deposited in the center of the plate in each assay plate. The plates were incubated at 30 °C for 7 days. After incubation, the growth of the fungal culture on fortified medium was evaluated. Those fungal plugs that showed no growth were transferred to fresh MEA plates without EO for an additional 7 days’ incubation at 26 °C to determine which concentration of each EO had a fungicidal effect. EO was considered fungistatic if growth resumed, otherwise it was considered fungicidal.

### 4.8. Vapor Contact Assay

The antifungal action of certain EOs in inhibiting the growth of *Aspergillus* spp. was also determined according to the inverted Petri Dishes Method [7,8,17,25,45]. The inoculum was represented by a 6 × 6 mm mycelium disc cut from the periphery of a seven-day-old aspergilli culture, which was deposited in the center of the plate in each test plate.

A paper disc (1 cm diameter) at dose levels of 1 μL/1 mL air space was laid on the inside surface of the upper lid and 50 µL of EO (5% and 75% *v*/*v*) was placed on each disc. Controls with DMSO and without EO were also prepared. The Petri dishes were sealed with parafilm and then instantly inverted on top of the lid to prevent the escape of EOs outside. Plates were incubated at 30 °C for 7 days. The effectiveness of antifungal activity of EOs was determined by assessing the presence or absence of fungal growth on the medium. In fungal cultures without visible growth, the lid was replaced to check the fungistatic or fungicidal action of the tested EO. Fungistatic activity was revealed by fungal growth after replacing the lid, whereas fungicidal activity was detected by the absence of growth on the agar plate.

### 4.9. Data and Statistical Analysis

The results were obtained from three independent experiments performed in triplicate and expressed as modal results. MICs and antifungal activity of AMB, ITZ and EOs against the tested fungal strains by disc-diffusion assay were compared using paired sample Student’s *t*-test (Graph Prism software, version 9 for Win-dows GraphPad Software, San Diego, CA, USA). The results were expressed as mean values, and *p*-values lower than 0.05 were considered significant.

## 5. Conclusions

Efficacy of EOs has been documented in several in vitro studies, which provide a basis for the activity of these natural compounds, but there are still issues to be addressed for their clinical use, including studies of pharmacokinetic, acute and chronic toxicity, mechanism of action and standardization of extraction techniques. The complexity of EOs is an obstacle to understanding their mechanisms of action and activity. When comparing the data obtained in different studies, the results obtained may differ. It is important to underline that the GC/MS qualitative analysis of the 11 essential oils tested highlights differences in composition and in particular as regards the main volatile compounds, which include two groups of distinct biosynthetic origin: terpenes/terpenoids and aromatics/aliphatic constituents, all characterized by low molecular weight [13,46,47]. Generally, these major components determine the biological properties of the EOs, and it is known that EOs are characterized by two or three major components even at high concentrations (20–70%) compared to other components present in traces [13]. Studies on the same EOs may give different results depending on the strains used and the constituents of the oils, and in some cases the constituents of the oils themselves may have a synergistic effect, so a variation in the percentages of the constituents may lead to different observed effects. The data highlighted in this work often confirm the effectiveness of many EOs, therefore these results can represent a valid starting point for the development of possible phytotherapeutic formulations designed to assist classic therapy or, alternatively, to reduce the use of drugs in patients especially debilitated by other diseases, where aspergillosis is in a secondary role.

## Figures and Tables

**Figure 1 molecules-28-07259-f001:**
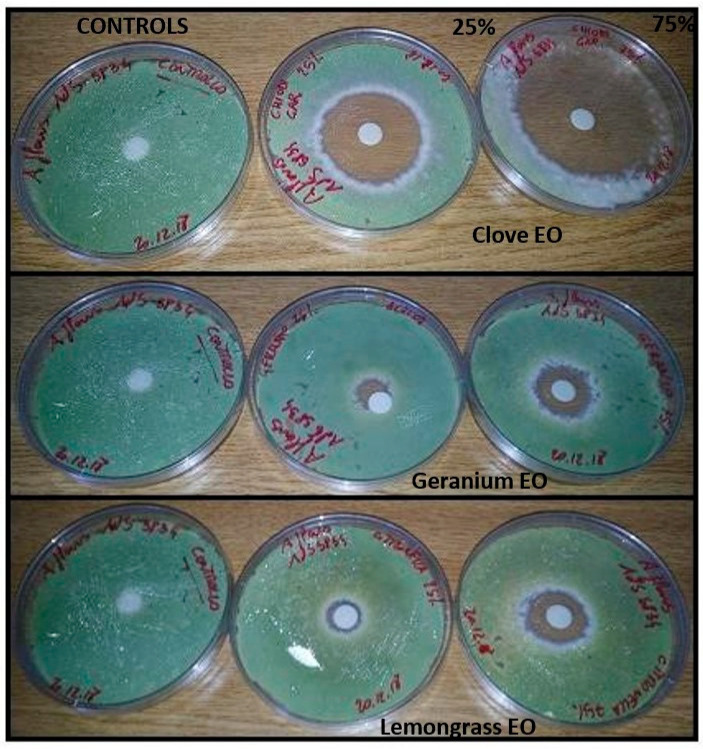
Inhibitory activity in agar medium of clove, geranium and lemongrass EOs 25% *v*/*v* (50 µg) and 75% *v*/*v* (150 µg) concentrations against *A. flavus* AdS 5834 strain. The diameter of the zone of inhibition of fungal growth around the disc corresponds to the effectiveness of the EO.

**Figure 2 molecules-28-07259-f002:**
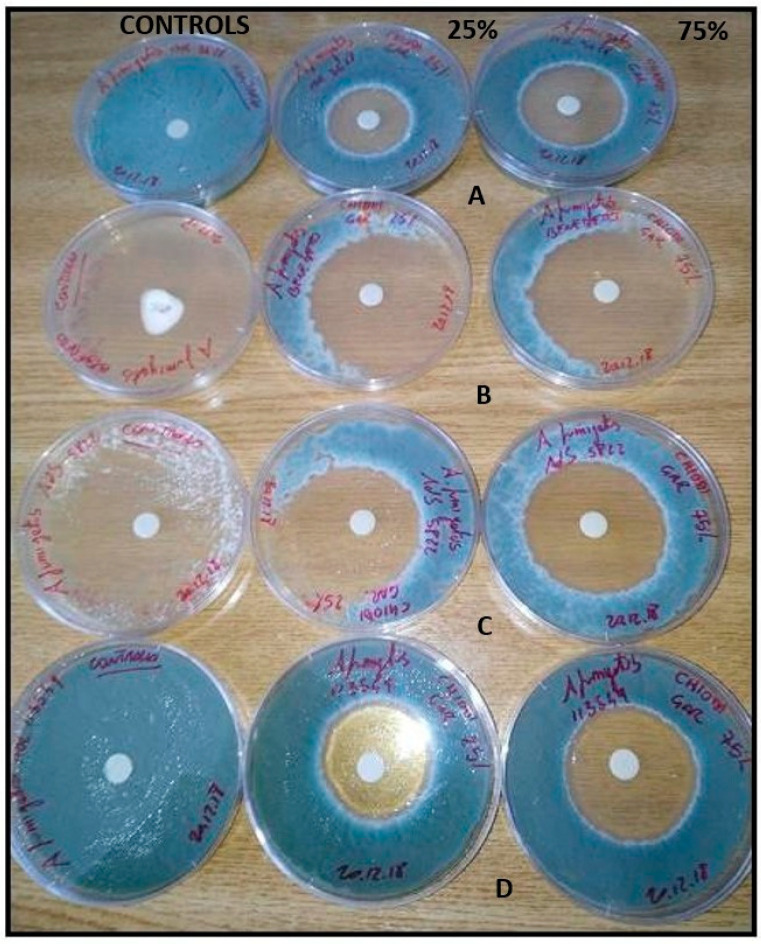
Inhibitory activity in agar medium of clove EO 25% *v*/*v* (50 µg) and 75% *v*/*v* (150 µg) concentrations against *A. fumigatus* MOL 3628 (A), BEN 177045 (B), AdS 5822 (C), MOL 113549 (D) strains. The diameter of the zone of inhibition of fungal growth around the disc corresponds to the effectiveness of the EO.

**Figure 3 molecules-28-07259-f003:**
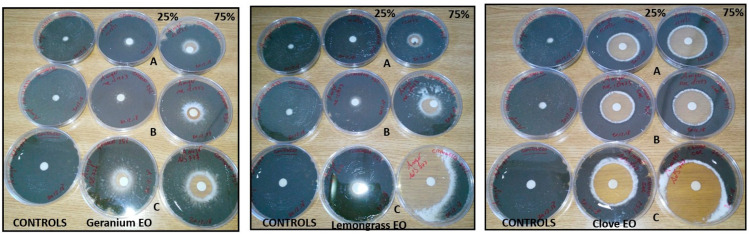
Inhibitory activity in agar medium of clove, geranium and lemongrass EOs 25% *v*/*v* (50 µg) and 75% *v*/*v* (150 µg) concentrations against *A. niger* AdS 121822 (A), MOL 121973 (B), AdS 777 (C) strains. The diameter of the zone of inhibition of fungal growth around the disc corresponds to the effectiveness of the EO.

**Figure 4 molecules-28-07259-f004:**
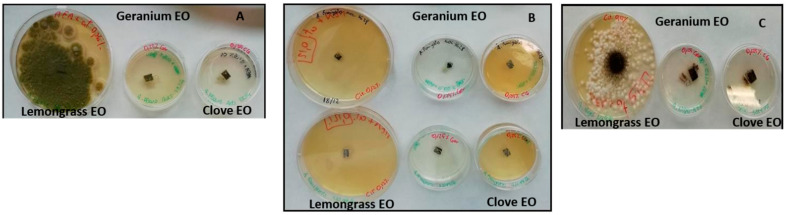
Test on medium supplemented with lemongrass, geranium and clove EOs against *A. flavus* AdS 5834 (**A**), *A. fumigatus* MOL 3628 and *A. fumigatus* 113549 (**B**), *A. niger* MOL 121973 (**C**) strains. EO was considered fungistatic if there was fungal growth, otherwise it was fungicidal.

**Figure 5 molecules-28-07259-f005:**
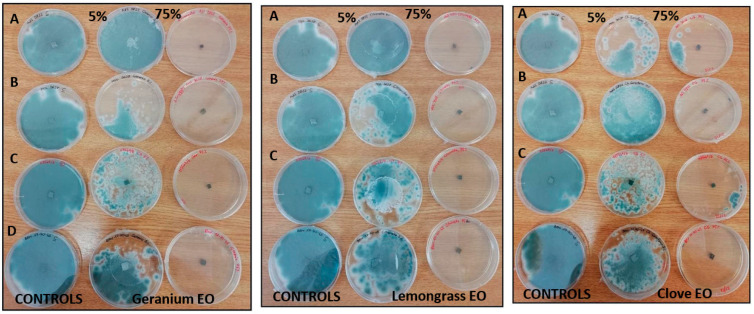
Vapor phase activity of geranium, lemongrass and clove EOs on *A. fumigatus* AdS 5822 (A), MOL 3628 (B), MOL 113949 (C), BEN 177045 (D). The effect of EO concentrations 5% *v*/*v* (10 µg) and 75% *v*/*v* (150 µg) was identified as fungistatic if growth was observed after the new incubation period, and fungicidal if no growth was observed.

**Figure 6 molecules-28-07259-f006:**
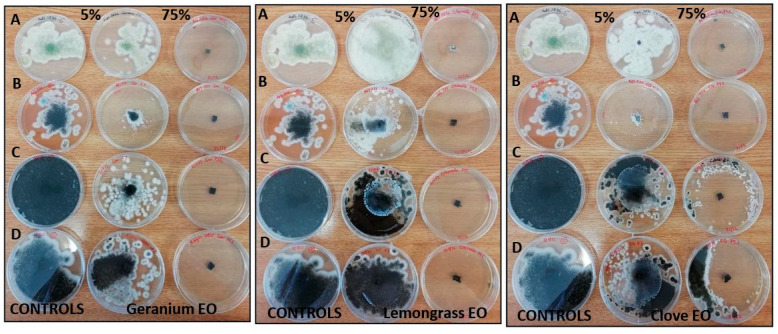
Vapor phase activity of geranium, lemongrass and clove essential oil on *A. flavus* AdS 5834 (A) and *A.niger* AdS 777 (B), MOL 121973 (C), AdS 121822 (D) strains. The effect of EO concentrations 5% *v*/*v* ( 10 µg) and 75% *v*/*v* (150 µg) was identified as fungistatic if growth was observed after the new incubation period, and fungicidal if no growth was observed.

**Table 1 molecules-28-07259-t001:** The eleven essential oils evaluated.

Family	Scientific Name	Common Name
*Lamiaceae*	*Lavandula x hybrid*	Hybrid lavender
	*Lavandula officinalis* P.	Lavender
	*Origanum vulgare* L.	Oregano
	*Thymus vulgaris* L.	Red tRed thyme
*Geraniaceae*	*Pelargonium graveolens* L’Hèrin.exAit.	Geranium
*Myrtaceae*	*Eugenia caryophyllata* Thumb.	Clove
	*Eucalyptus globosus* Labill.	Eucalyptus
	*Melaleuca alternifolia* Cheel.	Tea tree
*Poaceae*	*Cymbopogon nardus* L.	Lemongrass
*Rutaceae*	*Citrus lemon* L.	Lemon
	*Citrus aurantium var. bitter* L.	Neroli

**Table 2 molecules-28-07259-t002:** Minimum Inhibitory Concentration (MIC) of amphotericin B (AMB), itraconazole (ITZ), clove, eucalyptus, geranium, hybrid lavender, lavender, lemon, lemongrass, neroli, oregano, tea tree and red thyme EOs against the tested fungal strains.

Clinical Strains			MIC µg/mL			MIC EO mg/mL							
			AMB	ITZ		Clove		Eucalyptus		Geranium	Hybrid Lavender	Lavender	
*A. flavus*	AdS 5834		8	>16		2.25		>9		2.25	>9	>9	
*A. fumigatus*	AdS 5822		8	16		1.12		>9		0.56	>9	>9	
	MOL 4470		8	16		0.56		>9		2.25	>9	>9	
	MOL 4697		8	16		0.125		9		4.5	>9	9	
	MOL 1517		8	8		0.56		9		2.25	>9	9	
	BEN 177045		8	8		0.56		>9		0.56	>9	>9	
	MOL 113549		4	4		2.25		>9		2.25	>9	>9	
	MOL 3628		4	2		1.12		>9		1.12	>9	>9	
*A. niger*	AdS 777		4	4		0.56		>9		0.56	>9	>9	
	MOL 121973		2	4		1.12		>9		1.12	>9	>9	
	AdS 121822		2	4		0.56		>9		0.56	>9	>9	
	MOL 5015		4	2		1.12		>9		2.25	>9	>9	
	AdS 217		2	2		2.25		>9		2.25	>9	>9	
**Clinical Strains**			**MIC EO mg/mL**										
			**Lemon**		**Lemongrass**		**Neroli**		**Oregano**		**Tea tree**		**Red thyme**
*A. flavus*		AdS 5834	>9		0.56		>9		4.5		9		>9
*A. fumigatus*		AdS 5822	2.25		0.56		>9		>9		4.5		4.5
		MOL 4470	2.25		0.56		>9		>9		4.5		>9
		MOL 4697	>9		0.56		>9		>9		4.5		>9
		MOL 1517	2.25		0.56		>9		>9		4.5		9
		BEN 177045	2.25		0.56		>9		>9		4.5		9
		MOL 113549	4.5		1.12		>9		>9		4.5		>9
		MOL 3628	2.25		1.12		>9		>9		4.5		>9
*A. niger*		AdS 777	2.25		1.12		>9		>9		>9		>9
		MOL 121973	2.25		1.12		>9		>9		>9		>9
		AdS 121822	2.25		0.56		>9		>9		>9		>9
		MOL 5015	1.12		1.12		>9		>9		>9		>9
		AdS 217	4.5		1.12		>9		>9		>9		>9

**Table 3 molecules-28-07259-t003:** Minimum Fungicidal Concentration (MFC) of amphotericin B (AMB), itraconazole (ITZ) clove, geranium and lemongrass EOs against the tested fungal strains.

Clinical Strains		MFC µg/mL		MFC EO mg/mL		
		AMB	ITZ	Clove	Geranium	Lemongrass
*A. flavus*	AdS 5834	>16	>16	4.5	4.5	2.25
*A. fumigatus*	AdS 5822	8	16	4.5	2.25	2.25
	MOL 4470	8	16	2.25	2.25	4.5
	MOL 4697	8	>16	4.5	4.5	4.5
	MOL 1517	8	8	4.5	1.12	4.5
	BEN 177045	8	8	4.5	1.12	2.25
	MOL 113549	4	4	4.5	2.25	4.5
	MOL 3628	4	2	4.5	2.25	4.5
*A. niger*	AdS 777	4	4	9	2.25	2.25
	MOL 121973	2	4	9	2.25	2.25
	AdS 121822	2	4	9	4.5	2.25
	MOL 5015	4	2	9	4.5	4.5
	AdS 217	2	2	9	2.25	4.5

**Table 4 molecules-28-07259-t004:** Antifungal activity of clove, geranium and lemongrass EOs against the tested fungal strains by disc-diffusion assay.

Clinical Strains		Disc-Diffusion Assay (mm)							
		AMB	ITZ	Clove		Geranium		Lemongrass	
		20 µg	10 µg	50 µg	150 µg	50 µg	150 µg	50 µg	150 µg
*A. flavus*	AdS 5834	10 ± 0.1	14 ± 0.0	30 ± 0.0	66 ± 0.2 ^a^	13 ± 0.5 ^c^	22 ± 0.0 ^c^	12 ± 0.0 ^c^	21 ± 0.0
*A. fumigatus*	AdS 5822	12 ± 0.5	-	38 ± 0.2	42 ± 0.2 ^a^	12 ± 0.2 ^c^	40 ± 0.1 ^c^	13 ± 0.0 ^c^	38 ± 0.1
	BEN 177045	12 ± 0.1	25 ± 0.2	50 ± 0.1	55 ± 0.5 ^a^	29 ± 0.1	45 ± 0.0	14 ± 0.1	32 ± 0.0
	MOL 113549	11 ± 0.0	21 ± 0.1	29 ± 0.1	42 ± 0.0 ^a^	16 ± 0.0	25 ± 0.0	21 ± 0.2	51 ± 0.0
	MOL 3628	-	17 ± 0.0	26 ± 0.2	39 ± 0.0	14 ± 0.0	42 ± 0.4	19 ± 0.1	35 ± 0.1
*A. niger*	AdS 777	13 ± 0.0	14 ± 0.4	35 ± 0.4	56± 0.1 ^ab^	10 ± 0.0 ^c^	22 ± 0.6 ^c^	-	45 ± 0.2
	MOL 121973	12 ± 0.1	11 ± 0.1	29 ± 0.3	42 ± 0.0 ^ab^	-	16 ± 0.1 ^c^	-	20 ± 0.0
	AdS 121822	-	-	27 ± 0.0	43 ± 0.3	-	15 ± 0.0 ^c^	-	14 ± 0.4

Data expressed as mean of duplicate standard deviation (SD) including the disc diameter (6 mm), all values were expressed as µg/disc, ‘-’ indicates fungal growth. Positive control: amphotericin B and itraconazole. 50 µg= 25% *v*/*v*, 150 µg = 75% *v*/*v*
^a^
*p* < 0.05 vs. AMB; ^b^
*p* < 0.05 vs. ITZ; ^c^
*p* < 0.05 vs. clove 150 µg.

## Data Availability

The data presented in this review are available on request from the Corresponding Author Narcisa Mandras. The data are not publicly available due to privacy.

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
