# Peer review of "In Vitro Antifungal Activity of Selected Essential Oils against Drug-Resistant Clinical Aspergillus spp. Strains"

_molecules, 2023, doi:10.3390/molecules28217259_

Round 1
Reviewer 1 Report
Comments to the Authors
The manuscript entitled “In vitro antifungal activity of selected essential oils against drug-resistant clinical Aspergillus spp. strains.” has been reviewed. The topic is quite innovative and interesting, the MS is acceptable in present form. As a simple suggestion, the authors should consider to use the scientific name of the plants for the abstract.
Author Response
As a simple suggestion, the authors should consider to use the scientific name of the plants for the abstract.
Thank you very much for the comment. We have included the scientific name of the plants in the Table 1 because we have a limited number of words in the abstract.
Reviewer 2 Report
As a whole, the study provides information about the antifungal activity of various essential oils against drug-resistant clinical Aspergillus spp. strains. Lemongrass, clove, and geranium essential oils, in particular, demonstrated promising results and could be explored as potential alternatives or complementary therapies for managing aspergillosis. The research underscores the significance of natural products in combating fungal infections and highlights the need for further investigations in this area.
However, I have the following questions and comments for the authors:
I did not found the calculation of p-value to determine the statistical significance of differences between the samples. This make results for MICs and antifungal activity of AMB, ITZ and EOs against the tested fungal strains by disc-diffusion assay debatable.
The first paragraph from the "Discussion" repeat the information included in "Introduction" section and should be removed.
The quality of English language in the provided text is generally good, but there are some minor issues with grammar, punctuation, and sentence structure that could be improved for clarity and readability.
Author Response
ANSWER TO REFEREE 2 COMMENTS
I did not found the calculation of p-value to determine the statistical significance of differences between the samples. This make results for MICs and antifungal activity of AMB, ITZ and EOs against the tested fungal strains by disc-diffusion assay debatable.
We appreciated your suggestions. The results have been modified and now they are expressed as mean values and P-values using paired sample t-Student test (Graph Prism software for Window GraphPad Software. Please see Results section
The first paragraph from the "Discussion" repeat the information included in "Introduction" section and should be removed.
Following your suggestion, we deleted this part. Please see at page 8 in revised form. The revised part is marked with yellow highlight.
English Language
Done
Reviewer 3 Report
Dear editor,
Thank you for the invitation to review the manuscript “In vitro antifungal activity of selected essential oils against drug-resistant clinical Aspergillus spp. strains.”
After a thorough review, some points about the article should be clarified:
Introduction
Add more information about the antifungal potential of essential oils, as well as their possible mechanisms of action.
Results
1- I believe it is not possible to state the terms as a percentage of the essential oil in v/v. What is the difference between a chemical characterization carried out by GC/MS and GC/FID, and what is this relationship in the quantification of chemical constituents in essential oil?
2- In antimicrobial activity results, authors must standardize the concentrations of MIC found, in order to facilitate the authors' understanding µg /mL or v/v.
3- Do essential oils have antifungal or fungistatic potential similar to the antibiotics evaluated? How to compare?
Discussion
1- It needs to be improved by comparing data from the literature with the findings. Have the majority components of these oils evaluated already been tested against these pathogens? What are the MIC values? Is it compatible with the findings of this work?
Material and Methods
1- What is the antibiotic resistance profile of the clinical isolates used?
2- How was the chemical composition of essential oils created?
3- If the antifungal activity was performed according to the (CLSI M38-A2) protocol, how were the results described in %? Are there MIC values in µg/mL or mM?
The work was well written.
Author Response
ATTACHED THE ANSWERS TO THE REFEREE 3 COMMENTS

Round 2
Reviewer 2 Report
The paper is well-written, concise, and informative. The acceptance of recommended corrections further strengthens its quality. I recommend the publication of this article, as it contributes to the advancement of antifungal research and opens avenues for potential therapeutic alternatives.